# Prevalence of substance use and associated factors among secondary school adolescents in Kilimanjaro region, northern Tanzania

Rehema A. Mavura[1]*, Ahmed Y. Nyaki[1⊙], Beatrice J. Leyaro[2‡], Redempta Mamseri[1‡], Johnston George[1‡], James S. Ngocho[2‡], Innocent B. Mboya[1,2⊙]

**1** Community Health Department, Institute of Public Health, Kilimanjaro Christian Medical University College, Moshi, Tanzania, **2** Department of Epidemiology and Biostatistics, Institute of Public Health, Kilimanjaro Christian Medical University College, Moshi, Tanzania

⊙ These authors contributed equally to this work.
‡ BJL, RM, JG and JSN also contributed equally to this work.
* reyahmed1994@gmail.com

**Data Availability Statement:** The authors do not have a legal permission to share the data used in this study. All data requests should be directed to

## Abstract

### Background

Substance use among school-going adolescents increases the risk of developing mental disorders, addiction, and substance use disorders. These may lead to poor academic performance and reduced productivity, which affects adolescent lives. The study aimed to determine the prevalence of substance use and associated factors among secondary school adolescents in the Kilimanjaro region, northern Tanzania.

### Methodology

The study used secondary data from a cross-sectional survey of adolescents aged 10–19 years from public secondary schools in the Kilimanjaro Region, northern Tanzania. Substance use was measured using the Global School Health Survey (GSHS) questionnaire. Categorical variables were summarized using frequencies and percentages, while numerical variables used mean and standard deviation. Multivariable logistic regression models were used to obtain odds ratios (OR) and 95% confidence intervals (CI) to determine risk factors associated with lifetime and current (within the past 30 days preceding the survey) substance use.

### Results

The lifetime and current prevalence of substance use among 3224 adolescents was 19.7% and 12.8%, respectively, while alcohol and cigarettes were commonly used. Female adolescents had lower odds of current substance use (OR = 0.63, 95%CI 0.50–0.80). Higher odds of current substance use were among adolescents who have ever had sex (OR = 4.31, 95% CI 3.25–5.71), ever engaged in a physical fight (OR = 2.19, 95%CI 1.73–2.78), ever been bullied (OR = 1.55, 95%CI 1.16–2.05), always seen alcohol advertisements (OR = 1.87, 95%CI 1.37–2.53), and adolescents whose parent/guardians rarely understood their

the Institute of Public Health Director at KCMUCo through iph@kcmuco.ac.tz.

**Funding:** The author received no specific funding for this work.

problems (OR = 1.38, 95% CI = 1.03–1.85). Adolescents whose classmates always showed social support had lower odds of current substance use (AOR = 0.71, 95%CI 0.53–0.97). Similar factors were associated with lifetime substance users.

## Conclusion

The study reflects a high prevalence of substance use among adolescents in the Kilimanjaro region. Alcohol and cigarette are the most prevalent substances used. Regulatory measures are essential to limit alcohol advertisements that are media portrayed. Efforts are needed to reduce risk behaviors, such as physical violence and bullying, through peer support groups/clubs in school environments.

## Introduction

Substance use is the lifetime use of any substance, including khat (*Catha edulis*), cigarettes, illicit drug use, alcohol, and other substances [1, 2]. Substance use has increased in recent years and is a growing public health problem and a worldwide threat, significantly affecting young people aged 10–24 years [1–3]. The commonly used substances globally are alcohol, khat, cigarette, hashish, and other illicit drugs like cannabis, heroin, and cocaine [4, 5]. For instance, about 53% of people aged 15 years and above have ever used alcohol globally [6, 7]. A recent systematic review in sub-Saharan Africa estimated the prevalence of substance use among adolescents (10–19 years) to be 41.6%, with alcohol being the most prevalent (40.8%) compared to other substances [1, 2].

In Tanzania, the lifetime prevalence of substance use among school-going adolescents (11–17 years) was 7%, with alcohol at 4.5% and drugs (3.1%), specifically marijuana, amphetamines, or methamphetamines being the most used [8, 9]. In the Kilimanjaro region, cigarettes (15.5%), alcohol (9.2%), and marijuana (3%) were the most commonly used substances among school-going adolescents [10]. Studies on substance use and related risk behaviors among adolescents in Tanzania are scarce. Therefore, limited data is available to inform policy and interventions.

Adolescence marks a critical time of growth in the life course and profound changes in physical, cognitive, and social development [3]. Substance use at the early stages of adolescents increases the risk of developing addiction, mental disorders, and substance use disorders [1, 6]. For example, in developed countries, the estimated risk of developing drug dependence on cannabis uses alone among lifetime drug users who started using drugs during the adolescent period is 17% [11]. Substance use and mental disorders accounted for 183.9 million disability-adjusted life years (DALYs) in 2010, especially among adolescents and young to middle-aged adults aged 10–29 years [12–14]. In addition, substance use among school-going adolescents leads to poor academic performance, reduced productivity, high dropout rate, and indiscipline [4, 15], which have implications that can persist throughout the life course [16]. According to the Global initiative out-of-school children study in Tanzania, almost 2.3 million (57%) of secondary school-age children (14–17 years) are out of school; the reported dropout rate by 2014 among registered adolescents was 7.5% [17]. The factors are bad youth groups involved with substance use, specifically smoking bhang (marijuana) and truancy tendencies [17].

The Sustainable development Goals include strategies to reduce the burden of substance use among adolescents through strengthening the prevention and treatment of substance use [18]. Among other interventions, Tanzania's adolescent health and development strategy

2018–2022 aims to ensure the availability of preventive and treatment services that are afford-able, accessible, and friendly to reduce the burden of disease among adolescents. Also, the strategy recommends substance use counseling among adolescents, which would require pro-moting community-based youth centers and strengthening community involvement in the Adolescent Health Strategy (ADHS) to improve key adolescent health care practices [19]. Nev-ertheless, most existing interventions focus on sexual and reproductive health (SRH) and HIV/ AIDS and less on emerging adolescent issues such as substance use, accidents and injuries, mental health, and road safety [19].

In addition, studies on adolescent risk behaviors in different settings focused on one or a few substances, mainly involving young people aged 15–24 years and among out-of-school adolescents[6, 9, 16]. Therefore, information about the burden of substance use among the school-going adolescents (10–19 years) is relevant to complement the existing literature and inform targeted interventions and potential policy decisions. This study assessed the preva-lence of substance use and associated factors among secondary school adolescents in the Kili-manjaro region, northern Tanzania.

## Methods

### Study design, area, and population

We carried out a secondary analysis of data from a school-based cross-sectional study con-ducted in public secondary schools in four districts of the Kilimanjaro region, namely Moshi municipality, Moshi, Hai, and Siha districts, by the Institute of public health in Kilimanjaro Christian Medical University College (KCMUCO). The main aim was to assess the risk behav-iors of adolescents attending public secondary schools. The study included all consenting form-one students who attended public secondary schools in 2019 from the selected four dis-tricts of the Kilimanjaro region. Kilimanjaro is one of the regions in the Northern part of Tan-zania, comprising seven districts and covering 13250 Km$^2$. Kilimanjaro has an estimated population of 1,640,087 people and an annual growth rate of 1.6%. The major economic activi-ties in the region are agriculture and livestock keeping [20]. Kilimanjaro has many secondary schools compared to other regions, with 313 secondary schools (215 government, 98 private), making the region home to many adolescents who spend most of their time and days in school [21]. According to the country's profile, adolescent accounts for 23% of Tanzania's population, 13% and 10% for the 10–14 and 15–19 age groups, respectively [19].

### Sampling, data collection methods, and tools

A multistage sampling technique was used to select 3227 adolescents. Four districts were pur-posefully selected in the first stage, ensuring rural-urban representativeness. The second stage was a random selection of public secondary schools from all available schools in each district. Only form-one students were included at the school level to build a cohort of repeated cross-sectional surveys in the following years. The purpose was to document trends of adolescent risk behaviors over four years period (i.e., form-one to four). Finally, a simple random sam-pling technique selected students proportional to the size of each school. Analysis was per-formed on 3224 adolescents after excluding three participants (0.1%) aged less than ten years and greater than 19 years of age.

The interviews were self-administered, using the Regional School Health Survey (RSHS) questionnaire adopted from the WHO/CDC Global Student Health Survey (GSHS) [22]. The RSHS questionnaire was standardized to assess risk behaviors among school-going students in Tanzania and administered in the Kiswahili language [8]. The tool has also been extensively used in other settings[15, 23, 24]. The risk behaviors in this survey included nutrition and

participation in physical activity, personal hygiene, and substance use, including smoking, alcohol, illicit drugs, marijuana, khat use, experiences of violence and abuse, and risky sexual behaviors. Trained medical students from KCMUCo collected data. Before data collection, the data collectors explained the study purpose to all form one students and responded to all the questions asked before administering the interviews. The selected participants were then given the questionnaires and instructed to wait for further instructions from the data collectors. The next step was reading one question after another to ensure each respondent understood the question before filling out the questionnaire. The process continued until the last question. The data collectors made the necessary efforts to ensure privacy and confidentiality during the data collection. This was done by clearly explaining to participants why this was important and ensuring spaces between participants when completing the questionnaires.

## Study variables

The dependent variable in this study was substance use. Lifetime substance use refers to using any substance at least once throughout the adolescent's life [1, 2]. The substances considered in this study were alcohol, cigarette smoking, marijuana, khat, and recreational drugs (cocaine, heroin). Lifetime substance use was coded as "Yes" if an adolescent reported using any of the above substances and "No" if otherwise. Current substance use refers to using any of the following substances during the last 30 days preceding the interview: cigarette smoking, tobacco products, alcohol, recreational drug (cocaine and heroin), marijuana, khat, and amphetamines.

The independent variables included adolescent socio-demographic and behavioral characteristics. The demographic characteristics were adolescents age (10–14, 15–19 years), sex (male, female), schooling district (Moshi municipality council, Moshi district council, Siha, and Hai), and the number of days ever missed class (0 days, 1–3 days, ≥4 days). Social and behavioral variables were: parent/guardian use tobacco (neither, father or male guardian, mother or female guardian, both parents, don't know), ever had sex (no, yes), number of sexual partners (1 partner, ≥2 partners), source of alcohol (shop/ street vendor, gave someone else to buy, friends, family, stole, some other way), frequency seen alcohol advertisement (never, rarely, sometimes, most times, always), frequency parents/guardian understood your problems (never, rarely, sometimes, most times, always), number of close friends (no friends, 1–5 friends, >6 friends), social support from friends (never, rarely, sometimes, most times, always), ever been bullied (no, yes), ever engaged in a physical fight (no, yes), and ever rode in a car with a drunk driver (no, yes).

## Data processing and analysis

Data were cleaned and analyzed using SPSS software version 20. Descriptive statistics for substance use characteristics were summarized using frequencies and proportions for categorical variables. Continuous variables were summarized using mean and standard deviation. The Chi-square test determined the association between participant characteristics with lifetime and current substance use. Binary and multivariable logistic regression analysis estimated odds ratios (OR) and 95% confidence intervals (CI) for determinants of a lifetime and current substance use. A p-value of <0.05 was considered statistically significant in crude and adjusted analyses.

## Ethical consideration

The Kilimanjaro Christian Medical University College Research and Ethics Review Committee (KCMU-CRERC) approved the parent study. All people aged 18 years and above provided

oral informed consent. The headmasters from each secondary school assented to interview students aged <18 years because no invasive procedures required parental consent. Hence, the need for parental consent was waived by the ethics committee. Ethical approval for the current study was sought from the KCMU-CRERC and obtained approval ethical clearance certificate number PG04. The Institute Public Health director at KCMUCO provided permission to use the data. The study observed and protected the confidentiality and privacy of the subject's data. Instead of adolescent names or any personal identifiers, unique identification numbers identified study participants.

## Results

### Participant socio-demographic characteristics

The mean age of 3224 adolescents (10–19 years) in this study was 14.6 years and a standard deviation of 1.07 years. More than half (53.5%) were aged 10–14, and just over half (53.0%) were females. Most adolescents included in this study schooled in Moshi (41.6%) and Siha (21.2%) district councils and Moshi municipality (20.7%). Few respondents, 103 (3.2%), reported missing class at least for four days or more (Table 1).

### Self-reported prevalence of substance use

The lifetime and current prevalence of substance use was 19.7% and 12.8%, respectively. Among those who reported having ever used substances, the most common substance reported was alcohol (14.8%) and cigarette smoking (7.6%), followed by khat (1.6%), recreational drugs, specifically cocaine and heroin (0.7%), and marijuana (0.7%). The common substances reported among adolescents currently using substances were alcohol (8.2%) and cigarette smoking (4.3%) (Table 2).

### Adolescent social and behavioral characteristics

Among all the adolescents in this study, 13.5% had a father/male guardian in their family who was a smoker, compared to only 1.1% of mothers/female guardians. Among current alcohol

**Table 1. Participant socio-demographic characteristics (N = 3224).**

| Variables | Frequency | Percentage |
|---|---|---|
| **Age (years)** | | |
| Mean (SD) | 14.6 (1.07) | |
| 10–14 | 1726 | 53.5 |
| 15–19 | 1498 | 46.5 |
| **Sex** | | |
| Male | 1515 | 47.0 |
| Female | 1709 | 53.0 |
| **Schooling council** | | |
| Moshi Municipality | 667 | 20.7 |
| Moshi district council | 1342 | 41.6 |
| Hai district council | 533 | 16.5 |
| Siha district council | 682 | 21.2 |
| **Ever missed class** | | |
| Did not miss class | 2609 | 80.9 |
| 1–3 days | 512 | 15.9 |
| $\geq$ 4 days | 103 | 3.2 |

**Table 2. Self-reported prevalence of substance use (N = 3224).**

| Variable | Frequency | Percentage |
|---|---|---|
| **Ever use substances[†]** | | |
| Cigarette smoking | 246 | 7.6 |
| Alcohol | 477 | 14.8 |
| Recreational drugs [*] | 23 | 0.7 |
| Marijuana | 23 | 0.7 |
| Khat | 51 | 1.6 |
| **Lifetime (overall) substance use** | | |
| No | 2590 | 80.3 |
| Yes | 634 | 19.7 |
| **Current use of any substance [**][†]** | | |
| Cigarette smoking | 139 | 4.3 |
| Tobacco smoking | 40 | 1.2 |
| Alcohol | 264 | 8.2 |
| Recreational drugs [*] | 8 | 0.2 |
| Marijuana | 9 | 0.3 |
| Khat | 24 | 0.7 |
| Amphetamines | 51 | 1.6 |
| **Current use of any substance (overall)** | | |
| No | 2811 | 87.2 |
| Yes | 413 | 12.8 |

[*] Recreational drugs include both cocaine and heroin.

[**] Current substance use is within 30 days preceding the survey.

[†] Frequencies and percentages among those who answered "Yes".

users, most of them reported the source of alcohol was stealing (32%) and family members (30.8%). In addition, adolescents reported having always seen alcohol advertisements (15.3%), and few reported ever riding in a car with a drunk driver (4.5%) (Table 3).

On the other hand, almost ten percent of adolescents reported having ever had sex (9.3%), of which 58.5% had two or more sexual partners. Nearly two-thirds (65.6%) reported having 1–5 close friends, and 47.6% said their parents always understood their problems. More than one-third (36.9%) reported always getting social support from their classmates. The self-reported prevalence of bullying was 13.6%. Furthermore, less than a quarter reported having ever engaged in a physical fight (22.9%) (Table 3).

## Substance use by participant's social-demographic and behavioral characteristics

There were no significant differences in the proportions of a lifetime and current substance use by adolescent age groups. The proportion of lifetime substance use differed significantly by participant characteristics ($p < 0.05$). The proportion was higher among males (26.1%) compared to females (14.0%) and adolescents who missed class four days or more (29.1%) compared to those who did not (18.1%). In addition, the proportion of lifetime substance use was significantly higher among adolescents who: ever had sex (55.7), ever bullied (32.1%), ever engaged in a physical fight (32.8%), always seen alcohol advertisements (29.9%), and adolescents whose parents rarely understood their problems (25.0%) compared to their counterparts (Table 4).

**Table 3. Participant's social and behavioral characteristics (N = 3224).**

| Variables | Frequency | Percentage |
|---|---|---|
| **Parents use tobacco**[*] | | |
| Neither | 2588 | 80.3 |
| Father or male guardian | 436 | 13.5 |
| Mother or female guardian | 36 | 1.1 |
| Both parents | 16 | 0.4 |
| I do not know | 144 | 4.5 |
| **Source of alcohol in the past 30 days (n = 264)**[*] | | |
| From shop/street vendor | 13 | 5.2 |
| Gave someone else to buy | 10 | 4.0 |
| Friends | 52 | 20.8 |
| Family | 77 | 30.8 |
| Stole | 80 | 32.0 |
| Some other way | 18 | 7.2 |
| **Seen alcohol advertisements**[*] | | |
| Never | 1690 | 52.4 |
| Rarely | 1034 | 32.0 |
| Always | 493 | 15.3 |
| **Ever had sex** | | |
| No | 2924 | 90.7 |
| Yes | 300 | 9.3 |
| **Number of sexual partners(n = 300)**[*] | | |
| 1 partner | 118 | 41.5 |
| ≥ 2 partners | 166 | 58.5 |
| **Number of close friends**[*] | | |
| No friends | 288 | 8.9 |
| 1–5 friends | 2115 | 65.6 |
| >6 friends | 812 | 25.2 |
| **Received Social support from classmates**[*] | | |
| Never | 755 | 23.4 |
| Rarely | 1279 | 39.7 |
| Always | 1189 | 36.9 |
| **Parents/guardians understood your problems**[*] | | |
| Never | 822 | 25.5 |
| Rarely | 864 | 26.8 |
| Always | 1536 | 47.6 |
| **Ever been bullied** | | |
| No | 2785 | 86.4 |
| Yes | 439 | 13.6 |
| **Ever engaged in a physical fight** | | |
| No | 2486 | 77.1 |
| Yes | 738 | 22.9 |
| **Ever ridden in a car with a drunk driver** | | |
| No | 3079 | 95.5 |
| Yes | 145 | 4.5 |

[*]Frequencies and percentages do not add up due to missing values.

**Table 4. Substance use by participant characteristics (N = 3224).**

| Variable | Total (%) | Lifetime use n (%) | P-value | Current use [*] n (%) | P-value |
|---|---|---|---|---|---|
| **Age** | | | | | |
| 10–14 | 1726 (53.5) | 319 (18.5) | 0.07 | 213 (12.3) | 0.392 |
| 15–19 | 1498 (46.5) | 315 (21.0) | | 200 (13.4) | |
| **Sex** | | | | | |
| Male | 1515 (47.0) | 395 (26.1) | <0.001 | 261 (17.2) | <0.001 |
| Female | 1709 (53.0) | 239 (14.0) | | 152 (8.9) | |
| **District** | | | | | |
| Moshi MC | 667 (20.7) | 142 (21.3) | <0.001 | 93 (13.9) | 0.001 |
| Moshi DC | 1342 (41.6) | 310 (23.1) | | 201 (15.0) | |
| Hai | 533 (16.5) | 89 (16.9) | | 50 (9.4) | |
| Siha | 682 (21.2) | 93 (13.6) | | 69 (10.1) | |
| **Days missed class** | | | | | |
| Did not miss class | 2609 (80.9) | 473 (18.1) | <0.001 | 307 (11.8) | 0.001 |
| 1–3 days | 512 (15.9) | 131 (25.6) | | 86 (16.8) | |
| ≥ 4 days | 103 (3.2) | 30 (29.1) | | 20 (19.4) | |
| **Ever had sex** | | | | | |
| No | 2924 (90.7) | 467 (16.0) | <0.001 | 292 (10.0) | <0.001 |
| Yes | 300 (9.3) | 167 (55.7) | | 121 (40.3) | |
| **Ever been bullied** | | | | | |
| No | 2785 (86.4) | 493 (17.7) | <0.001 | 311(11.2) | <0.001 |
| Yes | 439 (13.6) | 141(32.1) | | 102(23.2) | |
| **Seen alcohol advertisement**[**] | | | | | |
| Never | 1690 (52.4) | 246 (14.6) | <0.001 | 152 (9.0) | <0.001 |
| Rarely | 1034 (32.0) | 241 (23.3) | | 165 (16.0) | |
| Always | 493 (15.3) | 147 (29.9) | | 94 (19.1) | |
| **Ever engaged in a physical fight** | | | | | |
| No | 2486 (77.1) | 392 (15.8) | <0.001 | 239 (9.6) | <0.001 |
| Yes | 738 (22.9) | 242 (32.8) | | 174 (23.6) | |
| **Parents/guardians understood your problems**[*] | | | | | |
| Never | 822 (25.5) | 246 (21.0%) | <0.001 | 152 (12.5) | <0.001 |
| Rarely | 864 (26.8) | 241 (25.0%) | | 165 (17.6) | |
| Always | 1536 (47.6) | 147 (16.0%) | | 94 (10.8) | |
| **Classmate social support**[**] | | | | | |
| Never | 755 (23.4) | 174 (23.1) | <0.001 | 113 (15.0) | 0.001 |
| Rarely | 1279 (39.7) | 272 (21.3) | | 182 (14.2) | |
| Always | 1189 (36.9) | 188 (15.8) | | 118 (9.9) | |

[*] Current use is within 30 days before the survey.

[**] Frequencies and percentages do not add up due to missing values.

Likewise, the proportion of current substance use was significantly higher among male adolescents (17.2%), those who missed class four days and more (19.4%), and those who ever had sex (40.3%) which was about four times higher than those who never had sex. Furthermore, the proportion of current substance use was significantly higher among those who had always seen alcohol advertisements (19.1%), ever engaged in a physical fight (23.6%), and whose classmate(s) never showed social support (15.0%) compared to their counterparts (Table 4).

## Adjusted analysis for factors associated with substance use

Significantly lower odds of lifetime substance use were among female adolescents compared to males (AOR = 0.61, 95%CI 0.50–0.74), those from Siha district (AOR = 0.64 95%CI 0.47–0.87) compared to Moshi municipal council, and whose classmates always showed social support (AOR = 0.73, 95%CI 0.57–0.95). Higher odds of lifetime substance use were among adolescents who have ever had sex (AOR = 4.50, 95%CI 3.60–6.13), ever been bullied (AOR = 1.50, 95%CI 1.17–1.94), ever engaged in a physical fight (AOR = 2.03, 95%CI 1.65–2.50), and always seen alcohol advertisements (AOR = 2.09, 95%CI 1.61–2.70) (Table 5).

**Table 5. Adjusted analysis for factors associated with substance use (N = 3224).**

| Variable | Lifetime substance use | | Current substance use | |
|---|---|---|---|---|
| | AOR† (95%CI) | P-value | AOR† (95% CI) | P-value |
| **Age** | | | | |
| 10–14 | 1.00 | | 1.00 | |
| 15–19 | 1.09 (0.70,1.32) | 0.378 | 0.97 (0.77,1.22) | 0.78 |
| **Sex** | | | | |
| Male | 1.00 | | 1.00 | |
| Female | 0.61 (0.50,0.74) | <0.001 | 0.63 (0.50,0.80) | <0.001 |
| **District** | | | | |
| Moshi MC | 1.00 | | 1.00 | |
| Moshi DC | 1.22 (0.96,1.57) | 0.107 | 1.14 (0.86,1.52) | 0.374 |
| Hai | 0.91 (0.66, 1.25) | 0.57 | 0.77 (0.52,1.14) | 0.193 |
| Siha | 0.64 (0.47, 0.87) | 0.005 | 0.78 (0.55, 1.12) | 0.180 |
| **Ever had sex** | | | | |
| No | 1.00 | | 1.00 | |
| Yes | 4.50 (3.60,6.13) | <0.001 | 4.31 (3.25,5.71) | <0.001 |
| **Days missed class** | | | | |
| Did not miss class | 1.00 | | 1.00 | |
| 1–3 days | 1.40 (1.09, 1.78) | 0.007 | 1.33 (1.00,1.76) | 0.049 |
| ≥ 4 days | 1.31 (0.80,2.14) | 0.287 | 1.09 (0.62, 1.94) | 0.75 |
| **Ever been bullied** | | | | |
| No | 1.00 | | 1.00 | |
| Yes | 1.50 (1.17,1.94) | 0.002 | 1.55 (1.16, 2.05) | 0.003 |
| **Seen alcohol advertisement** | | | | |
| Never | 1.00 | | 1.00 | |
| Rarely | 1.44 (1.16,1.79) | 0.001 | 1.50 (1.16, 1.93) | 0.002 |
| Always | 2.09 (1.61,2.7) | <0.001 | 1.87 (1.37, 2.53) | <0.001 |
| **Ever engaged in a physical fight** | | | | |
| No | 1.00 | | 1.00 | |
| Yes | 2.03 (1.65, 2.5) | <0.001 | 2.19 (1.73, 2.78) | <0.001 |
| **Parents/guardians understood your problems** | | | | |
| Never | 1.00 | | 1.00 | |
| Rarely | 1.16 (0.90, 1.49) | 0.253 | 1.38 (1.03,1.85) | 0.034 |
| Always | 0.81 (0.64,1.03) | 0.860 | 0.90 (0.68, 1.21) | 0.487 |
| **Classmate social support** | | | | |
| Never | 1.00 | | 1.00 | |
| Rarely | 0.89 (0.70, 1.13) | 0.327 | 0.89 (0.68,1.18) | 0.423 |
| Always | 0.73 (0.57, 0.95) | 0.017 | 0.71 (0.53, 0.97) | 0.029 |

† AOR: Adjusted Odds Ratio.

Furthermore, lower odds of current substance use were among female adolescents (AOR = 0.63, 95%CI 0.50–0.80) and whose classmates always showed social support (AOR = 0.71, 95%CI 0.53–0.97). The adolescents who ever had sex (AOR = 4.31, 95% CI 3.25–5.71), were bullied (AOR = 1.55, 95% CI 1.16–2.05), engaged in a physical fight (AOR = 2.19, 95% CI 1.73–2.78), always seen alcohol advertisements (AOR = 1.87, 95% CI 1.37–2.53), and whose parent/guardians rarely understood their problems (AOR = 1.38, 95% CI 1.03–1.85) had higher odds of current substance use (Table 5).

## Discussion

The study aimed to determine the prevalence and factors associated with substance use among secondary-school adolescents in the Kilimanjaro region, Northern Tanzania. The lifetime and current prevalence of substance use was 19.7% and 12.8%, respectively. The most prevalent substances used were alcohol and cigarette smoking. Factors significantly associated with life-time and current substance use included: sex (high among males), ever having sex, being bul-lied, ever in a physical fight, seeing alcohol advertisements, classmate's social support, and parents/guardians understanding adolescents' problems.

The lifetime prevalence (19.7%) of substance use among adolescents in this study is higher than 7% from the 2006 Tanzanian Global School-based Student Health Surveys (GSHS) [9]. Current substance use in this study is also higher than the 2017 GSHS, which reported specifi-cally alcohol (4.5%), drugs (3.1%), tobacco (5.1%), and cigarette smoking (4.5%) [8]. A study in Dodoma reported a higher prevalence of substance use, specifically alcohol (19.8%), smok-ing cigarettes (7.4%), and marijuana (3.3%) among adolescents aged 15–17 years [25]. The cur-rent prevalence (12.8%) of substance use in this study is lower than the WHO Global Alcohol status report on current use (21.4%) among adolescents aged 15–19 years in African regions [7] and a study done among adolescents in Ethiopia (47.9%) [26]. The possible explanation for the difference may be reporting only one substance in other studies, sample size variations, and adolescent age. Compared to the national 2006 and 2015 GSHS studies, the findings show a higher prevalence, possibly because of the small geographical coverage in this study. These findings demonstrate a need to strengthen regulatory measures to reduce substance use, par-ticularly alcohol and cigarette smoking.

We found no significant association between adolescent age and substance use. However, previous studies in Zambia [27] and Benin [28] in west Africa found high substance use prac-tice among adolescents 15+ years. Likewise, analysis of GSHS data from six Asian low- and middle-income countries revealed a higher risk of substance use (alcohol and smoking) among older compared to younger adolescents [29]. However, there are limited studies about the association between adolescent age and substance use in SSA. Despite the observed differ-ences with other studies, interventions should target reducing substance use practice among school-going adolescents because of their ingenuity in trying new things.

In this study, female adolescents had lower odds of using substances than males, similar to other studies in Morocco, Zimbabwe, and Ethiopia [26, 30, 31]. Alcohol use (among other sub-stances) is also common among sexually active adolescent males in SSA [32]. Lower odds of substance use among females may be associated with societal and cultural gender role expecta-tions to act and conduct themselves. Substance use among females is seen as a shameful, inap-propriate practice and less sensation-seeking behavior than males [26, 30, 31].

Adolescents who reported ever having sex were significantly more likely to be lifetime and current substance users. Likewise, in Tanzania, young people aged 15–24 years using alcohol were more likely to engage in risky sexual behaviors [6]. A study in SSA associated sexual behaviors with alcohol use [32]. The finding is also consistent with studies in Iran and

Bangladesh that associated the experience of sexual activity with substance use [24, 33]. The observed association may be because substances such as alcohol, cigarette, and illicit drugs affect cognitive processes and decision-making, thus contributing to a compromised judgment [24, 33]. Also, sexual activity during these periods can make adolescents vulnerable to developing mental disorders like depression or anxiety, which, in turn, could lead them to use substances [24]. Health education interventions on the effects and consequences of substance use should be enhanced in secondary schools [8].

As reported elsewhere, adolescents who have ever been bullied had higher odds of a lifetime and current substance use [34]. A possible reason for this is that the victims of bullying are predisposed to use substances as a maladaptive coping strategy [34]. In addition, a study in Thailand found that engaging in a physical fight was associated with alcohol use and misuse, where adolescents used substances as a coping mechanism [35]. Likewise, we also found that adolescents who engaged in a physical fight had a higher likelihood of substance use, similar to a study done across eight sub-Saharan countries [36]. Therefore, adolescents need to be educated on the psychological effects that bullying and physical fights can cause, explaining how victimization can cause severe depression and anxiety that lead to substance use [37]. Student support groups or systems for the affected may help the victims cope with the bullying and physical abuse, reducing dependence on substance use.

Adolescents who reported always seeing alcohol advertisements were more likely to use substances. Likewise, advertising alcoholic beverages in the mass media promoted abusive alcohol consumption in Italy [38]. Adolescents exposed to alcohol advertising are more likely to start consuming alcohol earlier and drink large amounts [39]. A systematic review also demonstrated the relationship between alcohol advertisements and increased consumption among adolescents [40]. These findings indicate the need to limit alcohol advertisements in the media or ensure they portray their adverse effects. They also suggest prohibiting selling, buying, and posting substances on school premises, especially among school-going and out-of-school adolescents [41].

This study's findings also show that adolescents whose classmates always showed social support were less likely to use substances. These findings are consistent with the study done in Bangladesh, which reported that the likelihood of substance use increases with a lack of peer support as it exhibits greater anti-social behaviors that can manifest in substance use [24]. In Malaysia, adolescents with inadequate peer support had a higher likelihood of substance use [42]. On the contrary, a study done in Ghana found no association between peer support and substance use [34]. The possible reasons could be that the study only assessed two substances, i.e., cannabis and amphetamines, compared to over five substances this study examined.

In addition, adolescents whose parents rarely understood their problems were more likely to use substances. Previous studies demonstrated that parents' supervision of adolescents reduces substance use practice [27, 43]. Limited parental monitoring, involvement, and active substance use in the home at the family level may predispose adolescents to use substances [44]. Therefore, lack of social support may expose adolescents to substance use practices as a coping mechanism due to insecurity.

## Study strengths and limitations

This study used a large sample size of 3224 adolescents from public secondary school schools in four districts of the Kilimanjaro region. Large sample size and wider geographical coverage enhance precision and study's representativeness, respectively. This study is one of the first in our setting to measure substance use's prevalence and associated factors. The study also estimated the burden of a lifetime and current substance use, specifically among adolescents aged 10–19 years, and associated factors, which is essential to inform necessary interventions.

The study had several limitations. Firstly, the study design is cross-sectional; hence cannot determine the temporal relationship between substance use and the associated factors. Secondly, the study collected data among adolescents attending public secondary schools. Thus, the results may not reflect students in private schools and out-of-school adolescents. Thirdly, the questionnaire used for data collection was adopted from the WHO/CDC Global Student Health Survey [22]. This tool does not capture all the factors associated with substance use. These factors include the place of upbringing, social sanctions, belief systems, self-control, cultural acceptance of substance use, and availability and accessibility of substances, particularly drugs and cigarettes/tobacco products[24, 26, 45]. The tool does not also capture the consequences of substance use among school-going adolescents. Lastly, there was also a possibility of recall and social-desirability bias where adolescents might have forgotten or answered what they thought was socially desirable. Such bias may either over or under-estimate the prevalence of substance use.

## Conclusion and recommendations

The study reflects the high prevalence of substance use among adolescents in the Kilimanjaro region. Alcohol and cigarette are the most prevalent substances used. The factors significantly associated with substance use were sex (high among males), ever having sex, being bullied, ever in a physical fight, seeing alcohol advertisements, classmate's social support, and parents/guardians understanding adolescents' problems. The study recommends that students' leadership, with support from the teachers, create support groups or clubs that may help adolescents share alternative healthy ways of coping/dealing with stress, anxiety, and depression caused by bullying, which leads them to rely on substance use. The government should adopt regulatory measures to limit the number of alcohol advertisements the media portrays. Monitoring adolescents' prohibition of selling and buying any substance such as alcohol and tobacco products is crucial in or near the school premises.

## Acknowledgments

The study was conducted as part of Master of Public Health training at KCMUCo. The authors acknowledge the Institute of Public Health at KCMUCo for permission to use the data used in this study. The authors also thank all medical students engaged in data collection and study participants whose responses enabled the availability of data used in this study.

## Author Contributions

**Conceptualization:** Rehema A. Mavura, Ahmed Y. Nyaki, Beatrice J. Leyaro, Redempta Mamseri, Johnston George, James S. Ngocho, Innocent B. Mboya.

**Data curation:** Rehema A. Mavura, Ahmed Y. Nyaki, Beatrice J. Leyaro, Innocent B. Mboya.

**Formal analysis:** Rehema A. Mavura, Ahmed Y. Nyaki, Innocent B. Mboya.

**Funding acquisition:** Rehema A. Mavura.

**Investigation:** Rehema A. Mavura, Ahmed Y. Nyaki, Innocent B. Mboya.

**Methodology:** Rehema A. Mavura, Ahmed Y. Nyaki, Beatrice J. Leyaro, Redempta Mamseri, Johnston George, James S. Ngocho, Innocent B. Mboya.

**Project administration:** Rehema A. Mavura, Ahmed Y. Nyaki, Innocent B. Mboya.

**Supervision:** Ahmed Y. Nyaki, Beatrice J. Leyaro, Redempta Mamseri, Johnston George, James S. Ngocho, Innocent B. Mboya.

**Validation:** Rehema A. Mavura, Ahmed Y. Nyaki, Beatrice J. Leyaro, Redempta Mamseri, Johnston George, Innocent B. Mboya.

**Visualization:** Rehema A. Mavura, Ahmed Y. Nyaki, James S. Ngocho, Innocent B. Mboya.

**Writing – original draft:** Rehema A. Mavura.

**Writing – review & editing:** Rehema A. Mavura, Ahmed Y. Nyaki, Beatrice J. Leyaro, Redempta Mamseri, Johnston George, James S. Ngocho, Innocent B. Mboya.

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
