## [Decision Letter · Decision Letter 0]

27 Apr 2022

PONE-D-21-31212Prevalence of substance use and associated factors among secondary school adolescents in the Kilimanjaro region, northern TanzaniaPLOS ONE

Dear Dr. Mavura,

Thank you for submitting your manuscript to PLOS ONE. After careful consideration, we feel that it has merit but does not fully meet PLOS ONE’s publication criteria as it currently stands. Therefore, we invite you to submit a revised version of the manuscript that addresses the points raised during the review process.

In particular, please address the comments regarding clarity in language and grammar. If you require assistance with this, we recommend that you utilize a fluent English speaker or a professional editing service; if you do so, please include in the cover letter of your manuscript the name of the colleague or professional editing service that assisted you.

We look forward to receiving your revised manuscript.

Kind regards,

Hugh Cowley

Staff Editor

PLOS ONE

Journal Requirements:

2. Please provide additional details regarding participant consent. In the Methods section, please ensure that you have specified (1) whether consent was informed and (2) what type you obtained (for instance, written or verbal). If your study included minors, state whether you obtained consent from parents or guardians. If the need for consent was waived by the ethics committee, please include this information.

5. Please amend the manuscript submission data (via Edit Submission) to include author Ahmed Y. Nyaki, Beatrice J. Leyaro, Redempta Mamseri, Johnston 

George, James S. Ngocho and Innocent B. Mboya.

Reviewers' comments:

Reviewer's Responses to Questions

**Comments to the Author**

1. Is the manuscript technically sound, and do the data support the conclusions?

Reviewer #1: Yes

Reviewer #2: Yes

2. Has the statistical analysis been performed appropriately and rigorously? 

Reviewer #1: No

Reviewer #2: Yes

3. Have the authors made all data underlying the findings in their manuscript fully available?

Reviewer #1: No

Reviewer #2: Yes

4. Is the manuscript presented in an intelligible fashion and written in standard English?

Reviewer #1: Yes

Reviewer #2: Yes

5. Review Comments to the Author

Reviewer #1: The authors of the manuscript; Prevalence of substance use and associated factors among secondary school adolescents in the Kilimanjaro region, northern Tanzania” sought to determine the prevalence of substance use and associated factors among secondary school adolescents in the Kilimanjaro region, Northern Tanzania. The authors report that the prevalence of substance use among school-going adolescents was higher than in previous studies conducted in Tanzania, with alcohol and cigarette us, being the most common substances used. They also noted sociodemographic and behavioral factors associated with substance use in this population. Overall, this manuscript is well written. However, there are some comments below for the authors to consider that would help improve the manuscript’s overall readability.

General comments: Please read entire manuscript thoroughly for typos and missing articles, incomplete sentences. I mentioned some below; but please cross-check the entire manuscript.

Abstract:

1. There is a typo in first sentence in the methods; missing “of”

2. Typo in second sentence in the methods; missing “were summarized”

3. “We used logistic regression to obtain odds ratios and 95% confidence intervals (CI) for risk factors associated with differences in PA”. What is PA?

4. “Multivariable logistic regression models was used to obtain odds ratios and 95% confidence intervals (CI)”. CI has been previously defined…

Background:

1. Typo in this sentence (missing aged): “Substance use has increased in recent years and is a growing public health problem and a worldwide threat, significantly affecting young people 10-24 years”

2. “For example, in developed countries, the estimated risk of developing drug dependence on cannabis uses alone…”; change uses to use

3. After looking it up, bhang seems to be marijuana? Can the authors clarify, many people in the international audience may not be familiar with the word..

Methods:

1. Please define KCMUCO

2. Please rewrite the following sentence; unclear as currently written: “A total of 3224 for the current study was analyzed data for adolescents after excluding three (0.1%) individuals aged less than ten and greater than 19 years.”

3. “Trained students of Doctor of Medicine to collect data.” Sentence is not complete.

4. “Lifetime substances use was defined as using any substance at least once in their lifetime”. Remove “s” from behind “substances”

5. Were the same substances examined for lifetime use, also examined for current use? I ask because I see amphetamines examined in current use, but not mentioned in lifetime use under the subsection “outcome and explanatory variables”. Can the authors clarify?

6. Did the authors consider source of the other substances examined besides alcohol as covariates? How did the authors select the independent variables to include in their study?

7. The authors can use one: crude or unadjusted

Results: I suggest that the authors redo or recheck the analysis they have presented in the results section; as well as the table arrangement and ordering.

1. I am not sure why the authors presented Table 3 before table 2 in their manuscript. Tables are supposed to be ordered…Table 1, Table 2….The authors should rearrange the way the results were presented or change the Table numbering so it’s sequential.

2. Can the %s come after the corresponding sentences/statements as the previous sentences? “…few adolescents (15.3%) reported having always seen alcohol advertisements and (4.5%) had ever rode in a car with a drunk driver.”

3. There are two table 2’s in this manuscript. Authors should correct this error. Also, are the percentages presented in the table for Substance use by participant characteristics, for the lifetime and current use row percentages or column percentages? Where are these numbers from? They are not adding up to 100. Can the authors clarify what they have done? Also, it would help if the authors put the n’s for the lifetime and current use in the table.

4. There are two table 3’s? Manuscript jumps from Table 4, in the results to Table 6, no mention of Table 5. Authors should arrange their results accordingly.

5. Last table: the authors mentioned in the methods that they presented crude/unadjusted and adjusted Cis…I do not see it presented in the tables. Can the authors clarify…

Discussion:

1. “The lifetime prevalence of substance use among adolescents in this study is high than the 2006 and 2017 Tanzanian Global School-based Student Health Surveys (GSHS)”. There is a typo in this sentence, change “high” to “higher”

2. When the authors are comparing prevalence to other studies in the discussion, they should remind the readers what prevalence they are comparing by providing the numbers they obtained in their study for comparison.

3. Can the authors provide more insight as to why they did not find a significant association between adolescent age and substance use in their current study?

3. Why is this sentence written this way with a period after the first sentence?: "These findings are consistent with the study done in Bangladesh. Which reported that the likelihood of substance use increases with a lack of peer support as it exhibits greater anti-social behaviors that can manifest to substance use"

Reviewer #2: TITLE

Prevalence of substance use and associated factors among secondary school adolescents in Kilimanjaro region, Northern Tanzania.

OVERALL COMMENTS

The study is an important contribution to the knowledge of the prevalence of substance use in Tanzania, specifically the Kilimanjaro region. Findings, especially factors related to substance use could influence policymakers in where to channel resources in dealing with the burden of substance abuse in the region. However, I recommend that the authors address these concerns below:

1. The study should mention some policies or efforts made to curb substance use amongst adolescents in Tanzania. It is worth mentioning if there is no policy at all. This could highlight the shortfalls in policies.

2. The study needs a conclusion to emphasize the results and clearly outline recommendations. It can also make suggestions for future studies.

3. The authors should consider using the assistance of a professional editor to correct grammatical and typing errors.

SPECIFIC COMMENTS

Abstract

1. The abstract has major grammatical errors that need to be addressed. I suggest authors use the assistance of a professional editor.

2. The statement in line 7, ‘...and whose parent/guardians rarely understood their problems’, is unclear and needs revision.

3. The sentence “The study used secondary data from a cross-sectional study adolescents aged 10- 19 years from public secondary schools in the Kilimanjaro Region, northern Tanzania” should be revised as “The study used secondary data from a cross-sectional survey of adolescents aged 10- 19 years from public secondary schools in the Kilimanjaro Region, northern Tanzania”

4. “Substance use was measured using the Global School Health Survey (GSHS) questionnaire.” – You should be as specific as possible.

5. What does “PA” stand for?

6. Similar factors [were] associated with lifetime substance users.

7. You indicated in the conclusion that “The prevalence of substance use among school-going adolescents in this study is higher than the previous studies in Tanzania …”. This conclusion is not directly derived from the study and should be deleted. Please note that your conclusion should exclusively be based on your results.

Background

In the last paragraph, the authors mention other studies focusing on only a few substances used by adolescents. Studies by Mnyika et al. in 2011, evaluated various substances used by adolescents in the region. I suggest authors rather focus on factors influencing substance use, as this was missing in other studies.

Methods

1. The last statement of the study design and population, ‘…According to the countries profile, adolescent accounts for 23% of Tanzania’s population, 13% and 10% for 10-14, 15-19 age groups,’ is unclear and need to be rephrased.

2. In the sampling data collection methods and tools, authors should consider rephrasing the first sentence of the second paragraph. Consider ‘Data was collected by medical students’, since the study has already been done.

3. Line 8 of the second paragraph of the sampling data, collection methods and tools should be revised. ‘the point ‘Trained medical students collected the data’ had been mentioned in line 1.

4. Authors should mention sources of stealing of substances as a limitation of the study and give future recommendations for other studies.

5. Table 2, statistics on number of sexual partners needs realignment.

Discussion

1. Authors need to provide the 2006 and 2015 GSHS statistics that were compared to the study.

2. The study, done in Ghana that found no association between peer support and substance use, needs to be cited.

3. Authors were able to compare each of the significant factors that were associated with adolescent substance use in the area to existing literature. However, they failed to compare the prevalence of other recreational substances like khat and methamphetamines to that of existing literature. They only compared alcohol, cigarettes, and marijuana, but no discussion on the other recreational substances.

Strengths and Limitations

The study in 2011 by Mnyika et al., titled ‘Prevalence of and predictors of substance use among adolescents in rural villages of Moshi district, Tanzania’, assessed substance use amongst adolescents. Moshi District was included in this study. I suggest you revise this statement that says this study is the first to be done in the region. Consider saying it is the first that includes associated factors to substance use.

Conclusion

The authors concluded that ‘The study’s prevalence of substance use is higher than the previous 2006 and 2015 Tanzania Global School-based Student Health Surveys’. This inference does not relate to the study results. The findings of the study did not result in this conclusion. I suggest the authors make concluding statements based on the findings of this study.

References

Reference No. 1 has the name of main author missing from lists of authors.

Minor Comments

1. In the Methods (p.6), you used “outcome”, “explanatory”, and “dependent” variable interchangeably. However, these terms are qualitatively different. In your case, it involves “substance use”, so I will suggest that you should use “dependent” variable throughout the paper.

6. PLOS authors have the option to publish the peer review history of their article (what does this mean?). If published, this will include your full peer review and any attached files.

Reviewer #1: No

Reviewer #2: No

---

## [Author Response · Author response to Decision Letter 0]

30 May 2022

Response: We have revised the entire manuscript to make sure it meets the journal requirements. 

2. Please provide additional details regarding participant consent. In the Methods section, please ensure that you have specified (1) whether consent was informed and (2) what type you obtained (for instance, written or verbal). If your study included minors, state whether you obtained consent from parents or guardians. If the need for consent was waived by the ethics committee, please include this information.

Response: Information about the type of consent and assent added in the ethics consideration section. 

Response: The manuscript was prepared in a word document and not in LaTex. Please advise if the entire manuscript is required to be transferred to LaTex and be re-submitted. 

Response: Apologies for this confusion. The data have not been deposited in a repository. The authors do not have a legal permission to share the data used in this study. All data requests should be directed to the Institute of Public Health Director at KCMUCo through iph@kcmuco.ac.tz. 

5. Please amend the manuscript submission data (via Edit Submission) to include author Ahmed Y. Nyaki, Beatrice J. Leyaro, Redempta Mamseri, Johnston 

George, James S. Ngocho and Innocent B. Mboya.

Response: The manuscript submission has been amended to include all authors. 

Reviewer reports:

Reviewer #1: 

Overall, this manuscript is well written, however there are some comments below for the authors to consider that would help improve the manuscript’s overall readability.

Response: We thank the reviewer for the very positive comment. We have revised the manuscript as recommended by the reviewer following the details below. 

General comments: Please read entire manuscript thoroughly for typos and missing articles, incomplete sentences.

Response: The manuscript has been thoroughly revised to correct for grammatical errors.

Abstract

1. There is a typo in first sentence in the methods; missing “of”,

Response: Thank the reviewer for this comment, the typo has been corrected

2. Typo in second sentence in the methods: missing “were summarized”

Response: Thank the reviewer for this comment, the typo has been corrected

3. We used logistic regression to obtain odds ratio and 95% Confidence interval (CI) for risk factors associated with differences in PA what is PA,

Response: We are sorry for this confusion. This was a typo and it is corrected to read “We used logistic regression to obtain odds ratios and 95% confidence intervals (CI) for risk factors associated with substance use”.

4. “Multivariable logistic regression models was used to obtain odds ratios and 95% confidence interval CI”.CI has been previously defined.

Response: We acknowledge the comment, it has been addressed, and deleted the repeated sentence. 

Background

1 Typo in this sentence (missing aged). “Substance use has increased in recent years and is a growing public health problem and a worldwide threat, significantly affecting young people 10-24 years”

Response: We acknowledge the typo and have addressed it.

2 “For example, in developing countries risk of developing drug dependence on cannabis, uses alone changes uses to use

Response: Thank you for pointing out the typo, we have addressed it.

3. After looking it up, bhang seems to be marijuana? Can the authors clarify, many people in the international audience may not be familiar with the word.

Response: We acknowledge the comment, and we have addressed it by adding in brackets the word “marijuana”, it was written bhang because, the global school initiative out of school children study was done in Tanzania, which aimed to determine the factors to why most children are out of school or dropouts. Smoking marijuana was one of the factors and the local language used by many students was “Bhang”.

Methods

1. Please Define KCMUCo.

Response: KCMUCo has been stated clearly in the methodology. 

2. Please re write the following sentence: unclear as currently written. “A total of 3224 for the current study was analyzed fata for adolescents after excluding three (0.1%) individuals aged less than ten and greater than 19 years”.

Response: We acknowledge the typo and clarified the sentence to read, “A total sample of 3224 adolescents was analyzed for the current study, after excluding three participants (0.1%) that were aged less than 10 years and greater than 19 years of age”.

3. “Trained Students of Doctor Of medicine to collect data is not complete”.

Response: Thank you so much for the comment, we corrected it to read “Trained medical students from KCMUCo collected data”.

4. “Lifetime substance use was defined as using any substance at least once in their lifetime” remove “s” from behind “substances”.

 Response: Thank you for pointing out the typo, we have addressed it.

5. Were the same substance examined for lifetime use, and also examined for current use? I as because I see amphetamines examined in current use, but not mentioned in lifetime use under the subsection “outcome and explanatory variables”. Can the author clarify?

Response: We thank the reviewer for this comment. The data collection tool was adopted from the WHO/CDC Global Student Health Survey (GSHS). In the assessment tool, the substances assessed for lifetime use were alcohol, cigarette smoking, marijuana, khat, recreational drugs (cocaine, heroin) while for current use, it was cigarette smoking, tobacco products, alcohol, recreational drug (cocaine and heroin), marijuana, khat, and amphetamines.

6. Did the authors consider the source of the other substances examined besides alcohol as covariates? How did the authors select the independent variables to include in their study?

Response: As indicated above, the assessment tool used was adopted from the WHO/CDC Global Student Health Survey (GSHS). So, as one of the limitations, it only assessed sources of alcohol and no other substances. The variables in this study were chosen in reference to the conceptual framework (Ludick and P’Olak, 2016) and past studies that used almost the same assessment tool adopted from WHO.

7. The author can use one crude/unadjusted

Response: We thank the reviewer for this comment, we acknowledge and chose to use one of these two words.

Results

General comment: 

Results: I suggest that the authors redo or recheck the analysis they have presented in the results section; as well as the table arrangement and ordering

Response: We acknowledge the reviewer’s comment, and we have rearranged the tables and correctly numbered them to match the explanation in the results section.

1. I am not sure why the authors presented Table 3 before table 2 in their manuscript. Tables are supposed to be ordered…Table 1, Table 2….The authors should rearrange the way the results were presented or change the Table numbering so it’s sequential.

 Response: We acknowledge the reviewer’s comment, and we have rearranged the tables and correctly numbered them to match the explanation in the results section

2. Can the %s come after the corresponding sentences/statements as the previous sentences? “…few adolescents (15.3%) reported having always seen alcohol advertisements and (4.5%) had ever rode in a car with a drunk driver.”

Response: We thank the reviewer for this comment, we corrected the statements to read “In addition, adolescents reported having always seen alcohol advertisements (15.3%), and few reported to ever rode in a car with a drunk driver (4.5%)”.

3. There are two table 2’s in this manuscript. Authors should correct this error. Also, are the percentages presented in the table for Substance use by participant characteristics, for the lifetime and current use row percentages or column percentages? Where are these numbers from? They are not adding up to 100. Can the authors clarify what they have done? Also, it would help if the authors put the n’s for the lifetime and current use in the table.

Response: We acknowledge the reviewer’s comment, and we have correctly numbered the tables to match the explanation in the results section. The lifetime and current use row or column percentages don’t add up to 100% because, as we noted below table 2, that the “frequencies and percentages presented are among those who answered “Yes” to ever used or currently using any substance. The numbers for the lifetime and current use all add up to “N= 3224” which is on the first line on table 2.

4. There are two table 3’s? Manuscript jumps from Table 4, in the results to Table 6, no mention of Table 5. Authors should arrange their results accordingly.

Response: We acknowledge the reviewer’s comment, and we have rearranged the tables and correctly numbered them to match the explanation in the results section

5. Last table: the authors mentioned in the methods that they presented crude/unadjusted and adjusted is…I do not see it presented in the tables. Can the authors clarify?

Response: We thank the reviewer for this comment. We have corrected the typo, as it was aimed to only present the adjusted analyses results. This was to reduce having many tables in this manuscript, hence it sufficed to only report the adjusted analysis results.

Discussion.

1. “The lifetime prevalence of substance use among adolescents in this study is high than the 2006 and 2017 Tanzanian Global School-based Student Health Surveys (GSHS)”. There is a typo in this sentence, change “high” to “higher”

 Response: Thank the reviewer for this comment, the typo has been corrected

2. When the authors are comparing prevalence to other studies in the discussion, they should remind the readers what prevalence they are comparing by providing the numbers they obtained in their study for comparison.

Response: We thank the reviewer for this comment, we acknowledge the inputs and changed the statements to include the prevalence of the other studies to remind the reader of what exactly we comparing.

3. Can the authors provide more insight as to why they did not find a significant association between adolescent age and substance use in their current study?

Response: We would like to clarify this that we did find an association between adolescents age and substance use in our current study, but the association was not statistically significant (P value was greater than 0.05), due to a little difference in the frequencies of the ages 10-14 (n= 1726 (53.5%) and 15-19 (n=1498 (46.5%)).

4. Why is this sentence written this way with a period after the first sentence? "These findings are consistent with the study done in Bangladesh. Which reported that the likelihood of substance use increases with a lack of peer support as it exhibits greater anti-social behaviors that can manifest to substance use"

Response: We acknowledge the reviewer’s comment and corrected that statement to read” These findings are consistent with the study done in Bangladesh, which reported that the likelihood of substance use increases with a lack of peer support as it exhibits greater anti-social behaviors that can manifest to substance use”.

Reviewer #2:

Overall comments, the study is an important contribution to the knowledge of the prevalence of substance use in Tanzania, specifically the Kilimanjaro region. Findings, especially factors related to substance use could influence policymakers in where to channel resources in dealing with the burden of substance abuse in the region. However, I recommend that the authors address these concerns below:

 Response: We sincerely appreciate the reviewer for the very positive comment. We hope the manuscript can be accepted for publication after addressing the reviewer and editorial comments.

1. The study should mention some policies or efforts made to curb substance use amongst adolescents in Tanzania. It is worth mentioning if there is no policy at all. This could highlight the shortfalls in policies.

Response: we acknowledge the reviewer’s comment, though we did mention in the background section, that the intervention and development strategies presenting the country mainly address adolescent issues most focus on HIV and sexual and reproductive health (SRH). We will include the lack of policy as well that address substance issues among adolescents.

2. The study needs a conclusion to emphasize the results and clearly outline recommendations. It can also make suggestions for future studies.

Response: we thank you the reviewer for this comment, we did change the conclusion and included few recommendations.

3. The authors should consider using the assistance of a professional editor to correct grammatical and typing errors.

Response: We thank the reviewer for this comment. The manuscript has been thoroughly revised for grammar. 

Abstract

1. The abstract has major grammatical errors that need to be addressed. I suggest authors use the assistance of a professional editor.

Response: We acknowledge the reviewer comment. The manuscript has been thoroughly revised for grammar.

2. The statement in line 7, ‘...and whose parent/guardians rarely understood their problems’, is unclear and needs revision.

Response: Thank you for this comment, we have addressed it. The statement now reads “adolescents whose parent/guardians rarely understood their problems”

3. The sentence “The study used secondary data from a cross-sectional study adolescents aged 10- 19 years from public secondary schools in the Kilimanjaro Region, northern Tanzania” should be revised as “The study used secondary data from a cross-sectional survey of adolescents aged 10- 19 years from public secondary schools in the Kilimanjaro Region, northern Tanzania”

 Response: we acknowledge the comment and have addressed it.

4. “Substance use was measured using the Global School Health Survey (GSHS) questionnaire.” – You should be as specific as possible.

Response: Details about the measurement of substance use in the GSHS are provided in the first paragraph of the study variables section.

5. What does “PA” stand for?

Response: We acknowledge the comment/ This was a typo as well, and it is corrected to read “We used logistic regression to obtain odds ratios and 95% confidence intervals (CI) for risk factors associated with substance use”.

6. Similar factors [were] associated with lifetime substance users.

Response: Thank you for pointing out the typo, we have addressed it.

7. You indicated in the conclusion that “The prevalence of substance use among school-going adolescents in this study is higher than the previous studies in Tanzania …” This conclusion is not directly derived from the study and should be deleted. Please note that your conclusion should exclusively be based on your results.

Response: we thank you the reviewer for this comment, we did change the conclusion to be based on our study results, which reads “The study reflects high prevalence of substance use among adolescents in Kilimanjaro region. Alcohol and cigarette are the most prevalent substances used. Regulatory measures are essential to limit alcohol advertisements that are media portrayed. Efforts are needed to reduce risk behaviors that such as physical violence and bullying, through peer support groups/clubs in school environments.”

Background

In the last paragraph, the authors mention other studies focusing on only a few substances used by adolescents. Studies by Mnyika et al. in 2011, evaluated various substances used by adolescents in the region. I suggest authors rather focus on factors influencing substance use, as this was missing in other studies.

Response: We thank the reviewer for this comment and pointing out the work by Mnyika et al (2011). However, our mentioning of “studies in other settings” was not specific to only those studies conducted Kilimanjaro region. Nevertheless, although Mnyika et al (2011) focused on alcohol and cigarette smoking, the study population was primary school students (standard 6 and 7) compared to form one (secondary school) students in this study. Our analysis included alcohol and tobacco use, among other substances. However, we acknowledge the point raised by the reviewer on identifying the factors associated with substance use, which was what we wanted to establish in this study. 

Methods

1. The last statement of the study design and population, ‘…According to the country’s profile, adolescent accounts for 23% of Tanzania’s population, 13% and 10% for 10-14, 15-19 age groups,’ is unclear and need to be rephrased.

 Response: Thank you for pointing out the typo, we have addressed it. The statements read, “According to the country’s profile, adolescent accounts for 23% of Tanzania’s population, 13% and 10% for the 10-14 and 15-19 age groups respectively” 

2. In the sampling data collection methods and tools, authors should consider rephrasing the first sentence of the second paragraph. Consider ‘Data was collected by medical students’, since the study has already been done.

Response: Thank you so much for the comment, we corrected it to read “Trained medical students from KCMUCo collected the data.”

3. Line 8 of the second paragraph of the sampling data, collection methods and tools should be revised, ‘the point ‘Trained medical students collected the data’ had been mentioned in line 1.

Response: Thank you for pointing out the typo, we have addressed it and deleted the repeated statement.

4. Authors should mention sources of stealing of substances as a limitation of the study and give future recommendations for other studies.

Response: Indeed, the tool did not capture a lot of other key factors that could explain substance use among adolescents in this study. We have highlighted such factors in the second paragraph of the study strengths and limitations and provided relevant citations. 

5. Table 2, statistics on number of sexual partners needs realignment

Response: we appreciated this comment which we would like to clarify. The number of sexual partners variables, the frequencies indicated was among those who responded yes to “ever had sex” that was (n=300) as indicated on the side of the variable, the * indicates there were missing frequencies of those who didn’t responded to how many sexual partners they had.

Discussion

1. Authors need to provide the 2006 and 2015 GSHS statistics that were compared to the study.

Response: We thank you the reviewer for this comment, we have addressed it and the statistics are provided in the statements which reads “The lifetime prevalence (19.7%) of substance use among adolescents in this study is higher than the 2006 Tanzanian Global School-based Student Health Surveys (GSHS) which was 7%. For current substance use in this study is also higher than the 2017 GSHS which reported specifically alcohol (4.5%), drugs (3.1%), tobacco (5.1%) and cigarette smoking (4.5%) but had no overall estimate. This information has been added at the beginning of the second paragraph of the discussion section. 

2. The study, done in Ghana that found no association between peer support and substance use, needs to be cited.

Response: Thank you for pointing out the typo, we have addressed it.

3. Authors were able to compare each of the significant factors that were associated with adolescent substance use in the area to existing literature. However, they failed to compare the prevalence of other recreational substances like khat and methamphetamines to that of existing literature. They only compared alcohol, cigarettes, and marijuana, but no discussion on the other recreational substances.

Response: Thank the reviewer this comment, we like to clarify. The overall prevalence of lifetime and current substance use was a composite measure of all substances and not specifically alcohol and cigarette smoking. The two substances, i.e., alcohol and cigarette were the commonly used among adolescents in this population which also agrees with the previous literature. In addition, the prevalence of other substances in this study had a combined prevalence of <5%.

Strengths and Limitations.

The study in 2011 by Mnyika et al., titled ‘Prevalence of and predictors of substance use among adolescents in rural villages of Moshi district, Tanzania’, assessed substance use amongst adolescents. Moshi District was included in this study. I suggest you revise this statement that says this study is the first to be done in the region. Consider saying it is the first that includes associated factors to substance use.

Response: We thank the reviewer for this comment, we have addressed and changed the statement. Which now reads “This study is one of the first in our setting to measure the prevalence and associated factors of substance use”

Conclusion.

The authors concluded that ‘The study’s prevalence of substance use is higher than the previous 2006 and 2015 Tanzania Global School-based Student Health Surveys’. This inference does not relate to the study results. The findings of the study did not result in this conclusion. I suggest the authors make concluding statements based on the findings of this study.

Response: We acknowledge the comments, we have addressed and changed the conclusion based on our findings.

Reference. 

No.1 has the name of the main author missing from lists of authors.

Response: Thank you for pointing out the typo we have addressed it.

Ogundipe O, Amoo EO, Adeloye D. Substance use among adolescents in sub-Saharan Africa : A systematic review and meta-analysis. 2018;2016. doi:10.7196/SAJCH.2018.v12i2.1524

Minor Comments

1. In the Methods (p.6), you used “outcome”, “explanatory”, and “dependent” variables interchangeably. However, these terms are qualitatively different. In your case, it involves “substance use”, so I will suggest that you should use “dependent” variable throughout the paper.

Response: we acknowledge the comment, we have addressed it and changed the sub-heading to 

“Study Variables.

---

## [Decision Letter · Decision Letter 1]

12 Jul 2022

PONE-D-21-31212R1Prevalence of substance use and associated factors among secondary school adolescents in Kilimanjaro region, northern TanzaniaPLOS ONE

Dear Dr. Mavura,

Thank you for submitting your manuscript to PLOS ONE. After careful consideration, we feel that it has merit but does not fully meet PLOS ONE’s publication criteria as it currently stands. Therefore, we invite you to submit a revised version of the manuscript that addresses the points raised during the review process.

We look forward to receiving your revised manuscript.

Kind regards,

Vanessa Carels

Staff Editor

PLOS ONE

Journal Requirements:

Reviewers' comments:

Reviewer's Responses to Questions

**Comments to the Author**

1. If the authors have adequately addressed your comments raised in a previous round of review and you feel that this manuscript is now acceptable for publication, you may indicate that here to bypass the “Comments to the Author” section, enter your conflict of interest statement in the “Confidential to Editor” section, and submit your "Accept" recommendation.

Reviewer #1: All comments have been addressed

Reviewer #2: (No Response)

2. Is the manuscript technically sound, and do the data support the conclusions?

Reviewer #1: Yes

Reviewer #2: Yes

3. Has the statistical analysis been performed appropriately and rigorously? 

Reviewer #1: Yes

Reviewer #2: Yes

4. Have the authors made all data underlying the findings in their manuscript fully available?

Reviewer #1: Yes

Reviewer #2: Yes

5. Is the manuscript presented in an intelligible fashion and written in standard English?

Reviewer #1: Yes

Reviewer #2: Yes

6. Review Comments to the Author

Reviewer #1: I thank the authors for addressing my concerns. The manuscript is much improved and should be received favorably by the target audience.

Reviewer #2: Thanks for being responsive to my reviews and suggestions. However, my remaining point is that the discussion should focus on studies conducted in Africa and other low- and middle-income countries. As such, the authors should consider revising the paragraph 3 of the discussion section that makes reference to studies conducted in the United States.

7. PLOS authors have the option to publish the peer review history of their article (what does this mean?). If published, this will include your full peer review and any attached files.

Reviewer #1: No

Reviewer #2: **Yes: **Hadii Mamudu

---

## [Author Response · Author response to Decision Letter 1]

25 Jul 2022

Reviewer comments:

1. My remaining point is that the discussion should focus on studies conducted in Africa and other low- and middle-income countries. As such, the authors should consider revising paragraph 3 of the discussion section that makes reference to studies conducted in the United States.

Response: Thank you for the good review and comment. We have edited paragraph three which states as follows: “However, previous studies in Zambia [27] and the United States[28, 29] found high substance use practice among adolescents 15 years and above. Likewise, analysis of GSHS data from six Asian low and middle-income countries revealed a higher risk of substance use (alcohol and smoking) among older compared to younger adolescents [30]. However, there are limited studies about the association between adolescent age and substance use in SSA.” 

References

27. Siziya S, Muula AS, Besa C, Babaniyi O, Songolo P, Kankiza N, et al. Cannabis use and its socio-demographic correlates among in-school adolescents in Zambia. Ital J Pediatr. 2013;39: 1–5. doi:10.1186/1824-7288-39-13

28. Jones CM, Clayton HB, Deputy NP, Roehler DR, Ko JY. Prescription Opioid Misuse and Use of Alcohol and Other Substances Among High School Students — Youth Risk Behavior Survey , United States , 2019. 2020;69: 38–46. 

29. Creamer MR, Jones SE, Gentzke AS, Jamal A, King BA. Tobacco Product Use Among High School Students — Youth Risk Data Source. 2020;69: 56–63.

---

## [Editor Report · Decision Letter 2]

9 Aug 2022

PONE-D-21-31212R2Prevalence of substance use and associated factors among secondary school adolescents in Kilimanjaro region, northern TanzaniaPLOS ONE

Dear Dr. Mavura,

Thank you for submitting your manuscript to PLOS ONE. After careful consideration, we feel that it has merit but does not fully meet PLOS ONE’s publication criteria as it currently stands. Therefore, we invite you to submit a revised version of the manuscript that addresses the points raised during the review process.

We look forward to receiving your revised manuscript.

Kind regards,

Hadii Mamudu, Ph.D

Guest Editor

PLOS ONE

Journal Requirements:

Additional Editor Comments:

Dear Dr. Mavura,

Thank you for revising your manuscript up to this point! However, you have not been responsive to the suggestion of the reviewer, i.e., "My remaining point is that the discussion should focus on studies conducted in Africa and other

low- and middle-income countries. ... the authors should consider revising paragraph 3 of the discussion section that makes reference to studies conducted in the United States." Nevertheless, you still make reference to studies in the United States where there has been enormous studies on substance use among youth and it will be under-reporting to refer to the United States in the Discussion. Indeed, I agree with the reviewer that the focus should be on studies conducted in African countries, not elsewhere.

While it is true that "there are limited studies about the association between adolescent age and substance use in SSA", extensive studies have been conducted on the use of substances such as tobacco using the GYTS data etc. The substances considered in this study are alcohol, cigarette smoking, marijuana, khat, and recreational drugs (cocaine, heroin); therefore, the discussion should relate to studies in SSA on any of these substances. GYTS studies on the association between age and initiation of tobacco use/cigarette smoking exist. As such, I will encourage the authors to review the literature and use the results to address the issue raised by the reviewer (including paragraphs 2 and 3). The GSHS does not capture the landscape of substance use among adolescents in SSA, so you should look outside that for such studies.

Thank you,

Prof. Hadii M. Mamudu

Guest Editor
---

## [Author Response · Author response to Decision Letter 2]

16 Aug 2022

1. My remaining point is that the discussion should focus on studies conducted in Africa and other low- and middle-income countries.

Response: Thank you for the good review and comment, we have edited paragraph we have edited paragraphs that did include cite studies from united states and other high income countries and focused more on sub-Saharan Africa ,low and middle income countries. The edited paragraph are will viewed on the document with track changes.

---

## [Editor Report · Decision Letter 3]

23 Aug 2022

Prevalence of substance use and associated factors among secondary school adolescents in Kilimanjaro region, northern Tanzania

PONE-D-21-31212R3

Dear Ms. Mavura,

Congratulation! We’re pleased to inform you that your manuscript has been judged scientifically suitable for publication and will be formally accepted for publication once it meets all outstanding technical requirements.

Kind regards,

Hadii Mamudu, Ph.D

Guest Editor

PLOS ONE

---

## [Editor Report · Acceptance letter]

25 Aug 2022

PONE-D-21-31212R3 

Prevalence of substance use and associated factors among secondary school adolescents in Kilimanjaro region, northern Tanzania 

Dear Dr. Mavura:

I'm pleased to inform you that your manuscript has been deemed suitable for publication in PLOS ONE. Congratulations! Your manuscript is now with our production department. 

Kind regards, 

on behalf of

Dr. Hadii Mamudu 

Guest Editor

PLOS ONE